# Ten Questions and Some Reflections about Palliative Care in Advanced Heart Failure Patients

**DOI:** 10.3390/jcm11236933

**Published:** 2022-11-24

**Authors:** Massimo Romano’

**Affiliations:** Organizing Committee Postgraduate Master in Palliative Care, University of Milan, 20122 Milan, Italy; max.romano51@gmail.com; Tel.:+39-3488131566

**Keywords:** palliative care, supportive care, heart failure, end of life care

## Abstract

Heart failure is a clinical syndrome with increasing prevalence, high morbidity and mortality. It is characterized by high symptom burden, poor quality of life and high economic costs. This implies that the heart failure (HF) patients who receive palliative care (PC) have needs similar to cancer patients, but which are often unmet. This paper analyzes the main unresolved issues regarding the relationship between HF patients and the referral to an early PC program. These issues are presented as ten questions related to which patients should be admitted to PC and at what stage of their disease. Furthermore, the barriers opposing to referral to PC, the role of cardiologists and PC physicians within the care team, the gap between the scientific societies’ suggestions and the real world, the right time to promote patients’ awareness and shared decision making, regarding prognosis, end of life wishes and choices, with reference also to cardiac implantable devices’ deactivation, are discussed. These unresolved questions support the need to reevaluate programs and specific models in achieving equal access to palliative care interventions for HF patients, which is still mainly offered to patients with cancer.

## 1. Introduction

Heart failure (HF) is a leading cause of death, with a high symptom burden in the advanced stage of disease and increasing economic costs. Hospitalizations are frequent [1].

For these reasons, HF patients frequently demonstrate palliative care (PC) needs, which are often unmet.

In this paper, ten important questions about this topic are posed, with the goal of deepening some related reflections.

## 2. Do Patients with Advanced Heart Failure (HF) Have a Need for Palliative Care (PC)?

Palliative care (PC), according to the World Health Organization (WHO), should be made available to all patients with special needs resulting from advanced, life-threatening diseases, including, but not limited to, cancer [2]. Each year, throughout the world, an estimated >52 million people (mainly those living in less developed countries) require PC, but fewer than 15% receive it [2]. WHO data from 2020 [2] revealed diagnoses of cancer in only 28% of adults > 20 years with documented PC needs.

The prevalence of Heart Failure (HF) worldwide varies from 1 to 3%, and increases with age [3,4].

The median incidence of HF is 2.2−3.2 [3].

HF is a leading cause of hospital admission, particularly in older patients [4,5].

Approximately one in three patients has been previously hospitalized for HF [6] and 50% are re-admitted within 1 year of their initial diagnosis of HF [7].

The number of HF hospitalizations is a strong predictor of cardiovascular (CV) and all-cause mortality [8,9].

Despite the efficacy of pharmacological and non-pharmacological treatments [10], the all-cause 1-year mortality rate remains high, up to 23.6% for acute HF and 6.4% for chronic HF [6] and up to approximately 50% at 5 years [2].

The prevalence of advanced HF, defined according to the European Society of Cardiology Position Statement for advanced HF [11], in a population-based cohort study of all Olmsted County, Minnesota, was 13.7% of patients enrolled from 2007 to 2017. The 1-year mortality was up to 49.9% and the time from diagnosis of advanced HF to death was 12.2 months [12].

Patients with advanced HF suffer from a significant symptom burden, similar to those with cancer, (dyspnea, pain, asthenia, depression, reduced spiritual welfare) as well as poor quality of life (QoL) (35–40% of patients) but they are assessed and treated less frequently [13]

Accordingly, patients with HF have important palliative care (PC) needs and their clinical consequence increases significantly in the last year of life [14].

## 3. What Do the Scientific Societies Suggest about PC in HF Patients?

The guidelines of the main international scientific societies [10,15,16,17] all refer to the need to start end-of-life (EOL) PC for patients with end-stage HF, to optimize their quality of life by providing adequate symptom control, in accordance with the wishes of the patients and their families.

However, the criteria proposed in the different guidelines to start PC in HF patients are not homogeneous.

The European Society of Cardiology (ESC) guidelines on advanced HF [11] suggest PC intervention at the EoL without specific criteria, with the goal to reduce symptom burden, to improve QoL and to define living will and advance directives, especially in patients with Mechanical Circulatory Support (MCS) or indication to Heart Transplantation (HT), according to the patient and family preferences.

A specific document of ESC [15] is dedicated to the implementation of PC in the treatment of HF. The document analyzes in detail when to initiate PC intervention, how to promote symptom identification and alleviation, how to support patient’s and their family’s spiritual and psychological distress and when and how to proceed to drug deprescription and cardiac implanted devices’ deactivation.

Frequent hospital admissions, need of palliative inotropic therapy, frequent Implantable Cardioverter Defibrillator (ICD) shocks, declining functional status and poor QoL are the main clinical criteria for the involvement of a PC physician.

The more recent Guidelines on HF of ESC [2] dedicate only one page to PC in advanced HF, according to the usual criteria of frequent hospital admissions over the last year, recurrent HF relapses, an NYHA class IV, a progressive decline in the QoL and non-eligibility to a HT or MCS.

The American Heart Association/American College of Cardiology (AHA-ACC) guidelines recommend: “For all patients with HF, palliative and supportive care-including high-quality communication, conveyance of prognosis, clarifying goals of care, shared decision-making, symptom management, and caregiver support-should be provided to improve QoL and relieve suffering” [16].

This assumption may be interpreted as a suggestion to start PC early in the HF patients’ trajectory, as pointed out by previous American Heart Association Statement [17] and the WHO Global Atlas of Palliative Care [18].

## 4. How Early Should the PC Be?

PC should be integrated as a support to disease-modifying therapies [18] every time in a patient with life-limiting or life-threatening disease, such as HF.

In fact, since the initial clinical phases, the patient with HF may suffer from distressing symptoms or psychological problems connected to the disease. However, there are not specific criteria defining the “true early period” to initiate PC.

Some criteria may be suggested to identify the right patient: advanced NYHA classes (III-IV), frequent hospitalizations (≥2 in the previous year), indication to implant ICD or Cardiac Resynchronization Therapy (CRT) [19].

Moreover, it should be mandatory to discuss prognosis and share decision-making.

In essence, the question is not if to start PC, but how.

In this context, the cardiological guidelines are elusive.

The old PC model, unfortunately still used, which is borrowed from oncology, concluded in the integration of a PC program only when the prognosis is poor for the next six months.

This model has been shown modest efficacy because the patient was often referred to PC late, when death was approaching [20]. The cardiologists do not pay attention specifically to a high symptom burden and low quality of life, since they focus on the issues related to their field of expertise, especially in the acute phases of the disease.

There are some recommendations toward a shift from interventions based on prognosis (so difficult to predict in the HF patient) to a model based on symptoms and quality of life improvement [17].

The most effective model of PC provides its early introduction, with a maintenance of the disease-modifying therapy (simultaneous care), gradually withdrawn according to the evolution of the underlying disease and the patient’s wishes. (Figure 1).

In advanced HF, the time may come when decisions concerning the full activation of PC or a program of MCS or HT should be taken [17]. At every phase of illness, new in-depth discussions with the patient, family and caregivers may be necessary, with the goal to reassess the shared decision-making related to the stage of disease, change in prognosis and therapeutic options. The discussion should be periodically updated, because HF often results in multiple relapses that nearly always result in hospitalization and, not rarely, lead to death.

## 5. Who Should Initiate PC in HF?

In this regard, it is also important to define the key expertise required for PC [21]. Primary competence is required for all healthcare professionals who take care of patients with severe diseases, clinical cardiologists obviously included.

These professionals should have knowledge and basic skills in PC. They should be able to assess and treat the most distressing symptoms (especially pain, anxiety, and depression).

Moreover, they should actively participate in the shared decision-making process, with in-depth discussion about the goals of treatment, patient preferences and advance directives related to end-of-life choices (including the possible need of cardiopulmonary resuscitation).

The specialist level includes healthcare professionals who are certified in the field of PC. Specialist PC consultations concern the treatment of refractory symptoms, spiritual and existential issues and the assistance required for the resolution of potential conflicts between family members and the attending team.

However, the different skills do not identify two separate times of care.

The cardiologists and PC doctors should work together, and the PC doctor should be an important component of the HF clinical network from the early phases of the HF [22].

A recent paper [23] analysed the efficacy of a specific model in which the PC specialist is early embedded in the HF team, taking particular care of patients with advanced HF, identified through indicators such as NYHA class III–IV, ejection fraction ≤ 0.30, hyponatremia or weight loss > 10%.

Moreover, the expert opinion has been also taken as the clue factor to refer patients with an estimated prognosis of 1–2 years.

This choice allows an “early” PC approach.

The model is based on a co-specialty team (HF cardiologist, HF geriatrician, HF nurse and PC physician), which allows patients to attend specialists together, at home, during hospital admission or at the outpatient clinic.

The results of this study are very interesting for some reasons:
More than 80% of patients reported to have felt listened toThe patients were more frequently enable to die at homeThe co-specialty model may ease the development of cultural contamination between the HF team and PC physician and consequently to overcome the barriers of early referral to PC.


## 6. How to Promote Awareness in HF Patients?

A key point in HF patients’ care is the patient awareness. The awareness should be linked to the knowledge of all components of the disease, including the indications and efficacy of pharmacologic and non-pharmacologic treatments, the unpredictability of the disease, with frequent hospital admissions, the role of additional comorbidities and poor outcomes, despite the optimal treatments.

The full prognostic awareness (PA) should be the result of this process of information, which is progressive, which means it is iterative and shared with the patient and family during the entire period of care [17].

Gelfman et al. assessed the prevalence of PA and Goals Of Care Discussion (GOCD) in 377 patients with advanced HF with an Implantable Cardioverter Defibrillator (ICD) [24]. The analysis was conducted from patients enrolled in the WISDOM trial, which evaluated the effect of multicomponent communication between patients and HF doctors regarding the ICD deactivation and advance care planning [25].

In the Gelfman study, PA was defined as a positive response to the question “Has your doctor ever told you whether you could die from your heart disease?”. GOCD was defined as positive response to the question “Have you and your doctor discussed any particular wishes you have about the care you would want to receive if you were dying?”.

Of the patients, 78% had PA, while among patients with PA only 26% had GOCD.

PA was inversely correlated to increasing age, whereas symptom severity was a positive predictor of GOCD.

The study showed the presence of a gap between PA and conversation regarding the choices at EoL, emphasizing the need to improve the physicians’ communication skills.

Discussions on advance care planning (ACP), requiring shared decision-making and including advance directives, are rarely offered to HF patients and families [17]. Cardiologists are not trained and rarely willing to discuss prognosis and end of life issues, due to cultural delay.

However, if it is most important to enhance the PA of patients, we have to remember that patients with serious chronic pathologies are sometimes more concerned with the fear of losing functional autonomy than their awareness of remaining life [26].

This means we need to identify the patient’s preferences regarding the prognostic information they wish to receive. This varies from patient to patient and in the different phases of disease. Together with the patient, the cardiologist should further explore this issue to improve patient information and awareness and that of the family/caregiver.

## 7. When and How to Discuss Prognosis with HF Patient and Family?

The goal of this paper is not to analyze and discuss the different prognostic scores, but to emphasize the general criteria of communication timing and modality to adequately inform the patient about the evolution of the disease.

Prognosis is a difficult and challenging issue to discuss for the physician, the patient and family members. It is a complex process involving the timing of communication (during the acute or stable stages of the illness), the uncertainty and unpredictability of the disease, the extent to which patient and family members want to know how long the patient has to live, the quality of the remaining life, the suffering due to the disease and the disproportionate invasive or aggressive treatments.

In essence, not only should the quoad vitam prognosis be considered, but also quoad valetudinem.

Uncertainty is a conditioning aspect in formulating a prognosis. The level of uncertainty is directly proportional to the level of complexity of the patient’s condition, linked to the specific characteristics of the primary disease, the number of comorbidities (cognitive impairment included) and to the interaction with non-biological factors (social, familial, financial, environmental and cultural status).

Even if it is difficult to live with uncertainty, physicians should accept it, understand it and know how to manage it. Knowledge of the limits imposed by uncertainty can make clinical judgment more reliable, in relation to uncertainty of prognosis in some cases and in relation to the decisions to adopt with end-of-life (EoL) patients [27].

Prognostic predictions are overall oriented to facilitate the timely adoption of eventual advanced treatments and/or palliative care; to choose appropriate and proportional treatments; to orient care choices towards the patient’s actual needs; to promote the freedom of personal choice; and to equally distribute the human and economic available resources.

This is the “four principles” approach to health care ethics: autonomy, beneficence, non-maleficence, justice [28].

## 8. What Are the Barriers to Implementing PC in HF Patients?

The number of patients who need PC differs substantially from the number who actually receive it, particularly in patients with cardiovascular diseases. Only 4% of HF patients in Great Britain receive PC, without significant changes over the years [29]. PC referral comes late for patients with HF, compared with cancer patients (12.4% vs. 30% respectively) [30].

Moreover, approximately 30% of these patients had been addressed to PC only during the last week of life.

What are the reasons for the gap between the practices advocated by the guidelines and those observed in clinical practice?

This is probably a cultural problem, linked to the belief that PC is indicated only for patients dying or suffering from cancer, or to fears that starting PC means all disease-modifying therapies will be discontinued. They embrace the tendency to “do everything”, even if inappropriate at the end of life. Furthermore, not all HF patients are being cared for by cardiologists (frequently geriatricians, specialists in internal medicine, general practitioners), and this raises the concern that PC might be started on the “wrong” patient, without full consideration of all available options.

Furthermore, short-to mid-term prognostic assessment is a challenging issue in patients with HF [10].

Moreover, the term “palliative care” is perceived as a stigma, linked to EoL, with a conceptual overlap between PC and EoL, although the non-synonymous nature of the terms has long been stressed by the WHO [2].

Some authors feel that the term palliative care may delay the early initiation of these measures, even for cancer patients.

Patients, family members and physicians consider PC as a care only for a patient at the EoL without hope and as provided in a hospice program.

The debate over the terminology and the role it plays in preventing physicians from starting PC has led to the proposal of the term “supportive care”, which seems preferred by many doctors and patients because it was not directly associated with death [31].

Finally, the lack of access to PC programs is often due to a shortage of PC specialist availability [22].

This is a serious problem, and in the future, it will be worse due also to 30% of physicians’ burnout and age (>56 or older) [32].

Lupu et al., while acknowledging the increased number of PC physicians in the US between 2010 and 2016, have predicted the need for PC specialists in the next 20 years will not be achieved, with a significant imbalance between the demand for PC physicians and supply [33].

## 9. How Many Patients with Advanced HF Receive Specialized PC?

The feasibility and effectiveness of PC intervention for patients with HF has been demonstrated in some Randomized Clinical Trials (RCT), reanalyzed in two different meta-analyses [34,35].

In the PAL-HF trial, 150 patients were randomized to usual care or to usual care plus PC intervention [36]. At 6 months of follow-up, patients randomized to usual care-PC showed significant improvement in quality of life, measured through the Kansas City Cardiomyopathy Questionnaire (KCCQ) overall summary and the Functional Assessment of Chronic Illness Therapy-Palliative Care scale (FACIT-Pal).

Moreover, depression and anxiety (measured via the Hospital Anxiety and Depression Scale (HADS)) and spiritual well-being (measured via the FACIT-Spiritual Well-Being scale (FACIT-Sp)) also showed significant improvement.

Patient-reported outcome measures included positive effects of palliative interventions on symptom burden and quality of life. Furthermore, the risk of at least one hospitalization was significantly reduced, without any effect on total mortality. Early initiation of PC allows a better control of symptoms, an improved QoL (for patients and their caregivers), an improved mood, an increased use of shared care planning and significant reductions in healthcare costs [34].

However, in clinical practice, HF patients are referred to PC programs, particularly by cardiologists (<15) quite late, at the end of life, with a very low Palliative Performance Status (PPS 0−30% in 29% of patients) [37].

The same results are shown in a study comparing hospitalized cancer patients and HF patients, admitted to a PC consultation.

The cancer patients were 3 times more frequently admitted to PC consultation in comparison with HF patients (30% vs. 12.4%, respectively), within a general population of 135.197 patients enrolled in the Palliative Care Quality Network in US [30].

Also, in this study, the HF patients, compared with cancer patients, were older (75.3 y vs. 65.2 y), had a significantly lower PPS (35.6% vs. 42.4%) and the referral location was more frequently a critical care unit (35.3% vs. 12.5%). This last point emphasizes the high burden of invasive and aggressive treatments up to the final days of life in HF patients.

Finally, a recent retrospective study in the Department of Veterans Affairs shows a very low incidence rate of HF patients who received PC, independent of ejection fraction (EF) (about 1 in 20 patients with reduced EF and 1 in 25 patients with mildly reduced or preserved EF) [38].

## 10. When Is the Right Moment to Discuss Cardiac Implantable Devices Deactivation at the EoL?

Discussions on advance care planning (ACP), requiring shared decision-making and including advance directives (AD), are rarely offered to HF patients and families [17]. Cardiologists are not trained and rarely willing to discuss prognosis and EoL issues, due to cultural delay.

HF patients are frequently implanted with electronic devices to prevent sudden death, such as with the Implantable Cardioverter Defibrillator (ICD), to treat left ventricular dysfunction and complete left bundle branch block through Cardiac Resynchronisation Therapy (CRT), eventually also associated with ICD (CRT-D), and to treat bradyarrhythmias with the Cardiac Pacemaker (PM).

In more advanced phases of the disease, they are likely to be supported by mechanical circulatory support devices (MCSDs), which include intra-aortic balloon pumps (IABP), percutaneous ventricular assist devices (e.g., Impella (Abiomed, Danvers, MA) and TandemHeart (LivaNova, Pittsburgh, PA)), extracorporeal membrane oxygenation (ECMO) and durable left ventricular assist devices (LVAD). LVAD may be indicated as a bridge to heart transplant (HTx) or as a destination therapy (DT) if HTx is not possible or contraindicated [11,16].

When the patient’s condition worsens at the final stages of life (concomitant multiple organ-failure, acute cerebrovascular events, cancer, chronic neurological diseases, dementias, sepsis, cardiogenic shock) the devices are unable to change the course of the disease.

The question may arise whether to deactivate these devices or to leave them active until the patient’s death [39,40,41,42].

The four ethical principles (autonomy, beneficence, non-maleficence and justice), in conjunction with the criteria of proportionality and appropriateness should lead the final decision [40].

In the case of LVAD as DT, the ethical concerns arise because these patients may develop LVAD complications (e.g., sepsis, bleedings, thromboembolic events) or other serious illnesses that may require palliative care or hospice care and subsequently the need to discuss the device deactivation.

This possibility should be discussed with the patient and family, as the ICD deactivation described below, in the pre-implantation phase, to improve their awareness [42].

In regards to the patients with ICDs, they might receive frequent and painful shocks from the device in the last stages of life.

The deactivation choice comes along with different clinical, ethical and relational issues specific to the different devices.

Cardiologists are generally late in dealing with these issues due to cultural, training and emotional reasons [43]. They feel great discomfort initiating discussion on treatment choices with patients and family members at EoL [40].

This compromises optimal patient care due to the inadequate attention to the quality of life and needs of advanced/terminally ill patients, particularly those with HF.

One reason for clinician discomfort with device withdrawal is the confusion between euthanasia or physician-assisted suicide and withdrawal of devices. Euthanasia is defined as an act, performed by a doctor, that intentionally causes death in someone who is very sick or suffering. In physician-assisted suicide, a physician helps a patient in taking his or her own life by intentionally providing a lethal dose of medication with knowledge that patient might commit suicide without active participation [42].

In device withdrawal, the physician’s intent is to remove the burdensome treatment and the cause of death is the underlying disease.

When dealing with devices deactivation, clinicians must comply with medical ethics principles, with special attention on the principle of patient autonomy.

Therefore, the discussion of the possibility of deactivating a device should start before device implantation, be part of informed consent and continue over time (continuous informed consent), with periodic updates in relation to the evolution of the clinical picture.

There is one more issue related to the ICD deactivation: Do Not Attempt Cardio-Pulmonary Resuscitation (DNACPR) orders. Cardio-Pulmonary Resuscitation (CPR) is an invasive therapy, not intended to be applied in patients in terminal phases of HF, due to its futility.

However, it is important to keep in mind that DNARCP orders do not mean do not treat/escalate [44].

Ideally DNACPR orders should join the patient’s decision of ICD deactivation.

However, discussions on ICD deactivation occurred in only 50% of patients who made comfort care and only 32% of patients who made DNACPR orders [45].

This implies that the patient needs to be fully informed about all end-of-life decisions, far from the terminal phases of life [45].

## 11. Should the Palliativist Be an Integral Part of HF Team?

To partially overcome the obstacles to promote early PC in HF patients there is a need for cultural contamination between the HF team and PC physician. The latter should be a permanent and early part of the HF team, supporting the discussion on ACP, EoL decisions, particularly about withholding/withdrawing life sustaining treatments [22].

Attempts to integrate PC into care plans for patients with HF have generally been unsuccessful for many reasons, such as the lack of validated prognostic criteria for starting PC, the absence of shared and validated models of care and the confusion related to the terminology [46].

There are two interrelated questions: what is the time for involvement of PC specialists and how to integrate them into the care team?

The decision to start PC in advanced HF patients should be based on the use of the risk scores specific for the disease, quality of life and decline/deterioration indicators, as PPS [47].

Early integration of a PC specialist into the HF care team is recommended by scientific societies, but it is rarely implemented [15].

The model shown in Figure 2 identifies two concentric rings around the patient: in the first one are included the professional figures who, at different times, may intervene in the patient’s care and contribute to their care planning.

The addition to the first ring of PC professionals allows for a more holistic management of the clinical, welfare, psychological and spiritual problems of the patient and his/her caregivers [34].

Secondly, the negative stigma surrounding PC professionals (especially in non-cancer clinical settings) could be reduced by including them as full members of the medical team. In this way, two parallel care plans could be defined from the beginning of disease (Figure 3) [29]. The first one, to implement specific disease-modifying therapies, the second one, addressed to identify the patient’s needs, as the assessment and management of symptoms, of psychological and spiritual distress, the discussion of prognosis and the process of shared care planning (supportive phase).

In the later stages, issues related to the implementation of planned treatments will be dealt with, including those destined for the most advanced stages of HF. This could be the beginning point of the palliative phase stricto sensu, when palliative inotropes may be used [5], and the EoL phase, when the withdrawal of life-supporting treatments, the deactivation of cardiac implanted electronic devices and the use of palliative sedation need to be addressed.

To improve the effectiveness of this model and to make it more widely applicable, there is the need for cultural contamination among the different physicians involved.

The PC team could approach the main issues in the clinical trajectory of HF patients, with particular attention to devices’ management (ICDs, PMs, LVADs) at the end of life.

Cardiologists might improve their attention to symptom assessment and treatment; moreover, they could start dealing with particularly difficult conversations.

Finally, this relationship should improve the process of shared decision-making in all stages of heart failure.

## 12. Conclusions

The problems analyzed in these ten questions are not all-encompassing of the relationship between heart failure patients and palliative care.

However, they represent the main unresolved issues that cardiologists and HF patients have to deal with.

The prevalence of palliative needs of HF patients is high, higher in older and in advanced HF patients: these needs are often unmet.

This supports the need to reevaluate programs and specific models in achieving equal access to palliative care interventions, which are still mainly offered to patients with cancer.

## Figures and Tables

**Figure 1 jcm-11-06933-f001:**
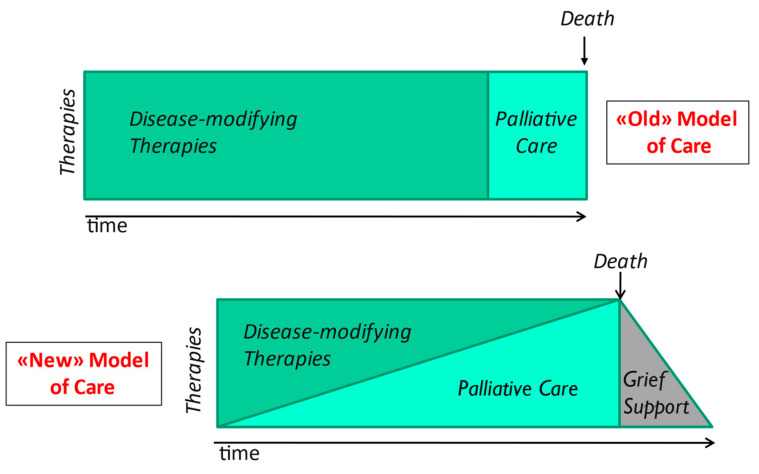
The different models to initiate palliative care.

**Figure 2 jcm-11-06933-f002:**
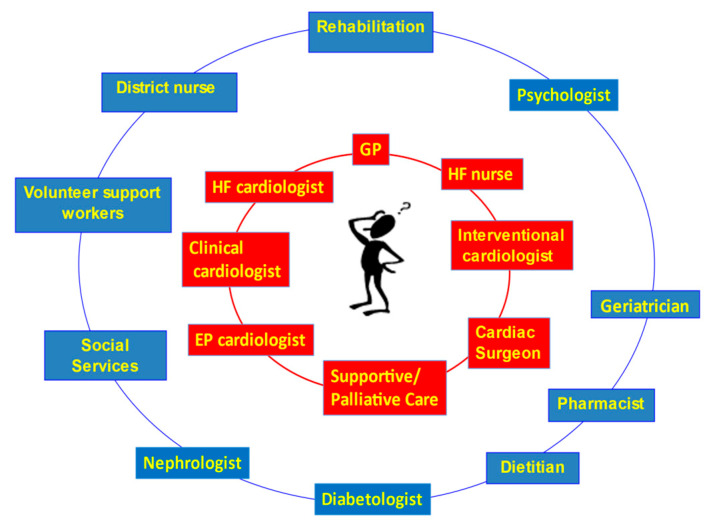
Heart Failure Clinical Network. The Palliative Care Physician is in the first ring surrounding the patient. GP: General Practitioner. EP: Electrophysiology HF: Heart Failure.

**Figure 3 jcm-11-06933-f003:**
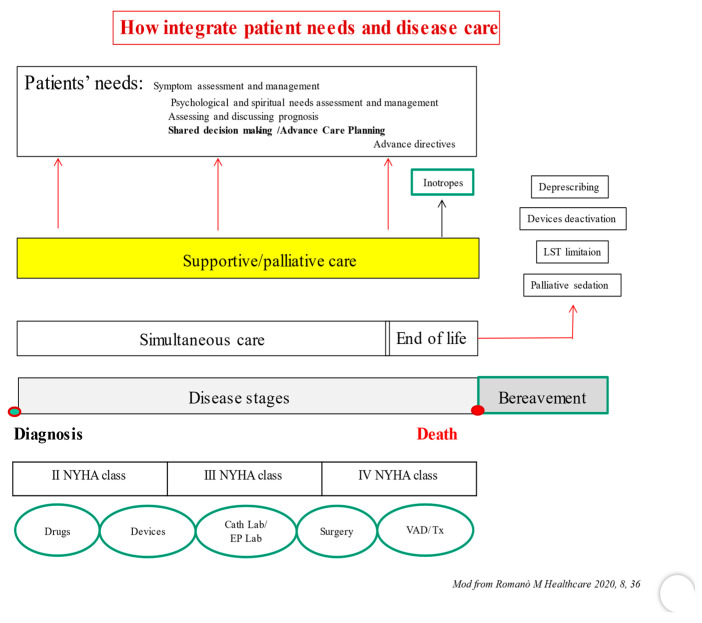
See text for details. VAD: Ventricular Assist Device, Tx. Cardiac transplant. EP Lab: Electrophysiologic Laboratory, Cath lab: Catheterization Laboratory, LST: Life Support Treatments.

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
