# Peer review of "Ten Questions and Some Reflections about Palliative Care in Advanced Heart Failure Patients"

_jcm, 2022, doi:10.3390/jcm11236933_

Round 1
Reviewer 1 Report
In this paper, Romano makes a complete review of palliative care in patients with heart failure, a current and completely necessary topic. In general, the manuscript is well structured and easy to read. I just add some comments and suggestions.
INTRODUCTION: Lines 29-34 repeat the same phrases as in the previous paragraph (23-28).
QUESTION 1: The reader probably already knows the epidemiological data of HF and its importance in terms of morbidity and mortality. They could summarize and emphasize the last paragraph (lines 64-71), which is what is really relevant to answer the question.
QUESTION 3: Perhaps the phases of the disease in which the palliative needs of patients should be evaluated could be highlighted, with the idea of ​​making a more practical review. Maybe this article could help the author: Slavin SD et al. Rev Esp Cardiol (Engl Ed). 2020;73:78-83.
Please, refer to any recent paper that shows that palliative care in HF patients is given very late, Line 117-118 (E.g. Fernandez-Martinez J et al. Int J Cardiol. 2021;327:125-31)
QUESTION 6: Cognitive impairment should be included among the determinants of HF prognosis.
QUESTION 7: Although I agree with most of the proposed barriers, I believe that the lack of availability of PC specialists should be emphasized. Undoubtedly the situation is not the same in all countries, but in many cases the shortage of specialists in this care means that they can only treat cancer patients.
QUESTION 8: I think the question should be modified: How many patients with advanced HF receive SPECIALIZED palliative care?
QUESTION 9: A paragraph could be added on do not resuscitate orders, an important topic related to ICD deactivation. Some previous studies have linked them to a deterioration in the quality of care and treatment for patients who have signed them (Cheng JLT et al. Am Heart J. 2008;156:76-84).Minor points:
Minor points:
In the manuscript, the cardiologist is identified as the doctor who treats patients with HF. However, the vast majority of these patients are managed by geriatricians, specialists in internal medicine or general practitioners. Moreover, it is this group of patients who most often have palliative needs, given that they are older and have multiple pathologies. This point should be clarified.
Author Response
I read with great attention the reviewers suggestions and I answered to all.
In detail:
Reviewer 1
- I deleted the lines 29-34 in Introduction
- Question 1: I summarized the epidemiological data (see lines 42-51), without a reduction of the main data.
- Question 3: I added the lines 110-112 and the new reference (n 19) and reference 20 (line121)
- Question 6: I added the lines 223-224.
- Question 7: I added the lines 273-280
- Question 8: I added the term Specialized in the question
- Question 9: I added the lines 369-379
Minor points: I added the lines 356-357

Reviewer 2 Report
Authors should be congratulated on this well written review. Comments
-Would expand the section dedicated to LVAD therapy.
-I did not see comments to :
ogers JG, Patel CB, Mentz RJ, Granger BB, Steinhauser KE, Fiuzat M, Adams PA, Speck A, Johnson KS, Krishnamoorthy A, Yang H, Anstrom KJ, Dodson GC, Taylor DH Jr, Kirchner JL, Mark DB, O'Connor CM, Tulsky JA. Palliative Care in Heart Failure: The PAL-HF Randomized, Controlled Clinical Trial. J Am Coll Cardiol. 2017 Jul 18;70(3):331-341. doi: 10.1016/j.jacc.2017.05.030. PMID: 28705314; PMCID: PMC5664956.
This would be an important trial to include in this report.
-Expand the role of the cardiologist in discussing goals of care.
Author Response
I read with great attention the reviewers suggestions and I answered to all.
In detail:
Reviewer 2
- I added the lines 338-346
- I added the lines 281-288
- Regarding the last suggestion I think it is analyzed in detail in the Question 5, dedicated to the awareness, goal of care and the role of the cardiologists. Moreover see the lines 351-353.
